# Role of Microorganisms in the Development of Quality during the Fermentation of Salted White Herring (*Ilisha elongata*)

**DOI:** 10.3390/foods12020406

**Published:** 2023-01-14

**Authors:** Jiajia Wu, Haiping Mao, Zhiyuan Dai

**Affiliations:** 1Institute of Seafood, Zhejiang Gongshang University, Hangzhou 310012, China; 2The Joint Key Laboratory of Aquatic Products Processing of Zhejiang Province, Hangzhou 310012, China

**Keywords:** salted herring, *Staphylococcus*, biogenic amine, volatile compounds

## Abstract

Salted white herring (*Ilisha elongata*) is a popular fish product in the coastal region of China. The complex endogenous enzymes and microbial action determine the quality of a traditionally salted herring. In order to investigate the role of microorganisms in the quality formation of salted herring, three groups for different salting processes were established: traditional salted (TS), non-starter salted (NS), and starter culture salted (SS). The predominant microorganism in each processing group was *Staphylococcus* spp., as inferred by next-generation sequencing data. Different physicochemical parameters were obtained in each of the three processing groups (TCA-soluble peptide (trichloroacetic acid-soluble peptide), TVB-N (Total volatile basic nitrogen), and TBA values (thiobarbituric acid-reactive substance)). The TS group had the maximum level of total biogenic amines, while the SS group had the lowest. A strong positive correlation was found between *Staphylococcus* and 14 aromatic compounds, of which 5 were odor-active compounds that created fishy, grassy, fatty, and fruity flavors. *Shewanella* may produce trimethylamine, which is responsible for the salted herrings’ fishy, salty, and deteriorating flavor. The findings demonstrated that autochthonous strains of *Staphylococcus saprophyticus* M90–61 were useful in improving product quality because they adapted quickly to the high osmotic environment.

## 1. Introduction

White herring (*Ilisha elongata*) is an essential commercial marine fish found in the Indo-Pacific region, including the coastal waters of China [1]. The white herring spawns from April to June when the water temperature rises above 19 °C [2]. In the past, fishermen would preserve the harvested white herring by heavily salting it. The consumption of salted white herring is popular in the coastal areas of China [3,4]. Processing techniques vary according to local food customs. The salted white herrings in Zhejiang Province are made from fresh whole fish, without scaling or gutting, and are salted three times: on day 0 (fresh fish), day 1, and day 5 [5]. The amount of salt used in the traditional method is greater than or equal to 40% of the wet fish weight. The unique flavor and texture of the salted herrings are normally formed after 1–3 months.

In the traditional salting method, a slender bamboo stick is impaled into the abdominal cavity through the gills up to the anus. The autolytic enzymes and microorganisms released from the digestive tract led to various biochemical reactions, which mainly defined the characteristics of the final product [6,7]. In traditional salted herring, the microbial community is both spontaneous and complex. The accumulation of biogenic amines and flavor compounds and the development of physical and chemical properties can be linked to the succession of microbiota [8].

Previous studies have revealed coagulase-negative staphylococci as the dominant genus in many different kinds of high salt-fermented seafood, such as *Saeu-jeotgal*, *Sanbao* large yellow croaker, *Budu*, and other fish sauces [9,10,11]. Autochthonous *Staphyloccocus* spp. can improve the development of flavor through proteolysis and lipolysis, followed by fatty acid β-oxidation, carbohydrate metabolism, and esterase activity [12]. However, undesirable microbiota or metabolites can also exist within this spontaneous complex fermentation, such as putrefactive bacteria, pathogenic microorganisms, or biogenic amine-producing bacteria [13]. Thus, research into the relationship between the microbial population and flavor development will aid in selecting starter cultures and regulating the quality of salted white herrings. Our previous study discovered that *Staphylococcus saprophyticus* was the dominant bacteria during salted white herring production. *S*. *saprophyticus* M90–61, a coagulase-negative *Staphylococcus*, was found to be a candidate for producing fermented salted white herring because it exhibited the following characteristics: it can grow even after being treated with 20% NaCl. It can grow well at 15–37 °C and produce protenase and lipase, even in 15% NaCl. Regarding safety, less biogenic amine was found in its culture medium, and no staphylococcal enterotoxin was found. It is resistant to common antibiotics. 

This study adopted an integrated process based on the Illumina MiSeq sequencing platform, analysis of physical and chemical properties, and GC-MS to investigate the microbiota succession, changes in physical and chemical parameters, and variation of the volatile components during the salting of herrings. Three different groups were set. It included a traditionally salted group (TS), a gutted but without starter culture group (NS), and a gutted group with *S. saprophyticus* M90–61 as the starter culture (SS). These groups helped reveal the influence of the gutting process and the addition of autochthonous starter culture on the microbiota composition and the development of quality in the final salted products. 

## 2. Materials and Methods

### 2.1. Preparation of Salted Herring Samples

Fresh white herrings were purchased from a local market in Hangzhou, Zhejiang Province, China. Three groups of salted herring were set. The traditional salted (TS) group was processed according to the traditional salting method. In this procedure, a long bamboo stick is impaled into the abdominal cavity through the gills up to the anus after thoroughly cleaning the fish with running water. Salt (NaCl 10% *w*/*w* wet fish) was applied on the surface of the fish and piled neatly with their abdomens upward for 24 h at 22 °C. During the second salting stage, half of the salt (NaCl 10% *w*/*w* wet fish) was stuffed into the abdomen, and the other half (NaCl 10% *w*/*w* wet fish) was applied to the surface. A pressure of about 15% of the fish weight was applied to the fish. Four days later, salt (NaCl 10% *w*/*w* of wet fish) was applied to the surface of the fish again, and the fish were kept under pressure (15% of fish weight) for 35 days. The non-starter salted (NS) group was processed based on the TS groups with a few modifications. Here, the gills and viscera were removed before pickling, and half the amount of salt was applied in each salting stage. The process of the starter culture salted (SS) group was similar to the NS group. Here, *Staphylococcus saprophyticus* M90–61 (isolated from the traditional salted herrings; data not shown) was inoculated (6–7 log CFU/g) on the 2nd, 5th, and 11th days of salting.

Samples from each group were picked on the 5th, 10th, 20th, and 40th day of salting and marked as TS5, TS10, TS20, TS40, NS5, NS10, NS20, NS40, SS5, SS10, SS20, and SS40, respectively. Fresh herring was also collected and marked as FH. The meat on the top back and about 0.5 cm behind the gills of the fish were collected and minced under sterile conditions. Three parallel groups were set for each sample. The samples were divided into small portions and stored at −80 °C.

### 2.2. Sequencing Analysis of the Bacterial Community

The microbial genome DNA was extracted using the E.Z.N.ATM Mag-Bind Soil DNA Kit (OMEGA, New York, NY, USA). Primer pairs of 341F (5′-CCTACGGGNGGCWGCAG-3′) and 805R (5′-GACTACHVGGGTATCTAATCC-3′) were used to amplify the V3-V4 region of the 16S rDNA gene [14]. The size and quantity of the amplicon library were assessed on the Agilent 2100 Bioanalyzer (Agilent, Santa Clara, CA, USA) with the Library Quantification Kit for Illumina (Kapa Biosciences, Woburn, MA, USA). The libraries were sequenced on the NovaSeq PE250 platform (San Diego, CA, USA) by LC-Bio Technology Co., Ltd. (Hangzhou, China). 

Paired-end reads were merged by FLASH (v1.2.11, http://ccb.jhu.edu/software/FLASH/index.shtml, accessed on 15 November 2022). High-quality clean tags were obtained after specific filtering on the reads according to fqtrim (v0.94, GitHub, San Francisco, USA). Chimeric sequences were filtered using Vsearch software (v2.3.4, GitHub, San Francisco, CA, USA). The feature table and sequence were obtained after dereplication using DADA2 [15]. OTUs clustering was analyzed by Uparse software (http://www.drive5.com/, accessed on 15 November 2022)., 

Alpha diversity was calculated by randomly normalizing to the same sequences. The SILVA (release 132) classifier was used to normalize feature abundance using the relative abundance of each sample. QIIME2 was used to calculate five alpha diversity indices, including Chao1, observed species, Goods coverage, Shannon, and Simpson, to analyze the complexity of species diversity. 

### 2.3. Determination of Physicochemical Parameters

Moisture content was calculated through the loss in weight of the sample after drying at 105 ± 2 °C for 8 h. Salt content was determined by silver nitrate titration as described by AOAC [16]. The pH of the sample was checked through the pH meter (Ultra Basic Benchtop, Arvada, CO, USA). Total volatile basic nitrogen (TVB-N) was determined through the semi-micro nitrogen method. Here, the distillate was collected, and the titration volume with a 0.1 mol/L HCl standard solution was calculated. The Kjeldahl method and extraction technique with petroleum ether were applied to measure the crude protein and crude fat content, respectively, according to the AOAC [16]. The trichloroacetic acid-soluble peptide (TCA-soluble peptide) was assessed through the procedure described by Wang et al. [17]. The thiobarbituric acid-reactive substance (TBA) values were measured by slightly modifying the method of Gao et al. [18]. The sample (10 gm) was homogenized (in 25 mL of 10% trichloroacetic acid and 25 mL of distilled water) for 5 min and then centrifuged (10 min at 5000 r/min). Then, 5 mL of the supernatant was mixed with thiobarbituric acid (5 mL of 0.02 mol/L) and incubated at 100 °C for 40 min. The absorbance was measured at 532 nm and 600 nm. The TBA value was calculated as follows: TBA value (expressed as mg malonaldehyde/kg sample) = (A_532_ − A_600_) × 72.06/155.

### 2.4. Determination of Biogenic Amines

Putrescine, cadaverine, histamine, octopamine, and tyramine were measured using the HPLC, according to the National Standard of the People’s Republic of China (GB/T 5009.208–2016) [19]. All the biogenic amine standards were purchased from Sigma-Aldrich (Merck, Munich, Germany).

### 2.5. Determination of Volatile Components

The volatile compounds were analyzed by the HS-SPME-GC/MS. Minced fish (5.0 g) was sealed in a 25 mL vial with a PTFE-lined cap. The volatile compounds were extracted at 60 °C for 40 min by the SPEM fiber coated with divinylbenzene/carboxen/polydimethylsiloxane (DVB/CAR/PDMS) film (50/30 µm, Supelco Inc., Bellefonte, AL, USA). 

The GC-MS (Gas Chromatography-Mass Spectrometer) (Thermo Trace DSQ II; Thermo Fisher Scientific, Waltham, MA, USA) analysis was carried out after desorption (250 °C for 4 min) with a TR-35 MS column (0.25 mm × 30 m, 0.25 µm; Thermo Fisher Scientific, Waltham, MA, USA). Helium (purity 99.999%) was the carrier gas (flow rate of 1.0 mL·min^−1^). The GC oven temperature was initially programmed at 40 °C for 2 min, then increased to 60 °C at a rate of 4 °C·min^−1^, then to 100 °C at a rate of 5 °C·min^−1^, and finally to 240 °C at 8 °C·min^−1^ for 6 min. The source and interface temperatures were set at 200 °C and 280 °C, respectively. The mass spectra were determined in EI+ mode at 70 eV within 30–500 *m*/*z* (mass-charge ratio). 

The volatile flavor compounds were identified by matching the mass spectra fragment with the NIST 2008 library (Hewlett-Packard Co., Palo Alto, CA, USA) and 2-, 4-, and 6-trimethylpyridine (500 µg/100 g) as the internal standard. 

The odor activity value (OAV) was calculated as follows:

OAV = C/OT, 

C is the concentration of VOC 

OT is its odor threshold (obtained from the literature [20]). 

### 2.6. Statistical Analysis

The experiment was repeated thrice for all samples, and the data were expressed as mean ± standard deviation (SD). A two-way analysis of variance (ANOVA) was used to determine significant differences, as the evisceration with inoculum (TF, NS, and SS) and processing time (5, 10, 20, and 40 days) were considered two independent factors. Duncan’s test was used to determine the statistical differences between data groups, and significance was set at *p* < 0.05. O2PLS modeling was conducted using Simca-14.1 software (Sartorius Stedim, Sweden) to reveal the relationship between the volatile flavor and different processing techniques. A correlation heat map was generated to exhibit the relationship between the volatile flavor accumulation and microbial abundance through R version 3.6.3. (R-Tools Technology, Inc., Richmond Hill, ON, Canada)

## 3. Results and Discussion

### 3.1. Microbiota Succession in Different Groups of Salted Herring

The variations in the bacterial community in the three different processing groups were reflected in the high-throughput sequencing data. The alpha-diversity indices assessed the richness and diversity of the bacterial community in salted herring (Table A1). The addition of salt reduced the abundance and diversity of microorganisms in the different salting groups when compared to fresh herring. The maximum diversity and abundance of microorganisms were observed in the TS group. A similar trend was observed in the NS group, but at a lower level. Although, the richness and diversity of microbiota in the SS groups dropped significantly after the 10th day of salting.

Figure 1 shows the phylum and genus of the microorganisms present. The dominant phyla in fresh herring were Proteobacteria and Bacteroidetes. In the TS5, TS10, NS5, and NS10 groups, the addition of salt reduced the Bacteroidetes while increasing the Proteobacteria. Since the 5th day of salting in the SS group and the 20th day of salting in the TS and NS groups, the Firmicutes have gradually increased and become the dominant phylum. 

The indigenous microbial genus of *Psychrobacter* (37.97%) and *Flavobacterium* (28.09%) were predominant in fresh herring, followed by *Acinetobacter* (6.14%), *Pseudomonas* (5.87%), *Shewanella* (5.26%), and *Pseudoalteromonas* (4.28%). An evident change in the microbial community in each processing group was seen. In the TS5 sample, *Photobacterium* (46.02%), *Vibrio* (20.62%), and *Shewanella* (16.90%) were in the majority. *Shewanella* (41.32%) outcompeted the other genus and became the predominant bacterial population in the TS10 sample. The population of *Staphylococcus* spp. started increasing from the 10th day of salting and became dominant in the TS20 (38.91%) and TS40 (58.65%) samples. A remarkable difference in the microbial flora of the NS group in comparison to the TS group was observed. *Vibrio* was predominant in NS5 (57.44%) and NS10 (38.74%), followed by *Psychrobacter* and *Pseudoalteromonas. *Staphylococcus** became the dominant strain in the NS20 and NS40 samples. Since the starter culture was added to the SS group on the 2nd day of salting, *Staphylococcus* spp. became the predominant bacterial strain on the 5th day of fermentation. The proportion of *Staphylococcus* was 93.41% in SS20 and 98% in SS40, which inhibited the growth of other genera. *Staphylococcus* gradually became the predominant population in each salting group. This result concorded with other studies, where the coagulase-negative staphylococci were predominant in salty fermented foods due to their tolerance to large amounts of salt [10,12,21].

The lack of gutting in traditional salted herring processing caused an initial increase in gram-negative spoilage bacteria such as *Shewanella* and *Photobacterium* before the *Staphylococcus* populations rose, even though a large amount of salt was used. The proliferation of the spoilage bacteria produced unpleasant odors and harmful metabolites, potentially threatening food safety [22,23]. The gutting conducted in the NS and SS groups reduced the proportion of *Shewanella*, thereby improving the safety of the final product. However, less use of salt and no starter culture induced the growth of *Vibrio* in the early salting stage of the NS group. The excessive use of salt in traditional salting inhibited the proliferation of pathogenic and spoilage bacteria effectively [24]. 

The isolation and characterization of halophilic bacteria or strongly halotolerant microorganisms was helpful in the standardization of the quality of fermented seafood [25], as these autochthonous cultures can quickly adapt to the environment and multiply into a dominant status. Similarly, in this work, autochthonous culture (*S. saprophyticus* M90–61) in the SS group multiplied fast and effectively inhibited the proliferation of spoilage and pathogenic bacteria.

### 3.2. Variations in the Physical and Chemical Parameters during the Salting of Herring

Figure 2 depicts variations in the physical and chemical properties of the three experimental groups, and Table A3 displays data obtained through a two-way Anova analysis. A similar trend in the variation of water and salt content was observed across the three groups after adding salt. Despite the fact that the TF groups used twice as much salt as the other two groups, there was no significant difference in the final salt content. The salt contents in the fish muscle were deemed to be saturated under current production conditions. The pH value of the TF group fluctuated, with a rebound on the 5th day and a gentle ascent after the 20th day of salting, which finally stabilized at 6.5. The pH values of the other two groups dropped steadily to 6.3–6.4. The pH value is affected directly by the microbial metabolites, endogenous enzymes, and catalytic products. 

As the moisture content decreased, a rapid increase in the protein content was observed in all three groups with fluctuation. The TCA-soluble peptide index was estimated to evaluate protein degradation and was significantly influenced by both the processing time and technique (Table A3). An initial rapid increase in the TCA-soluble peptide in the TF group was observed, which remained high till the end of the experiment. The viscera and head contain more autolytic enzymes than other tissues [6], which might increase protein degradation in the TF group. TVB-N in fish constitutes ammonia (NH_3_), dimethylamine (DMA), and trimethylamine (TMA), which is used to estimate spoilage or decomposition and is an index for the freshness of the fish [26]. TVB-N accumulated in each group due to amino acid degradation and was significantly influenced by the gutting processing (Table A3). Changes in TVB-N levels were similar in the NS and SS groups but significantly higher in the TF group, which increased slightly in the final salting phase (Figure 2). Endogenous and microbial enzymes are the primary causes of protein decomposition and TVB-N accumulation in the TF group [3,12].

There was a rapid increase in the fat content of all the samples in the initial salting phase due to the quick loss in moisture content. After that, a similar trend with fluctuations in the fat content was observed in each group. The processing technique had no significant influence on the fat content variations (Table A3). The TBA values were estimated to evaluate fatty acid oxidation, which is responsible for the off-flavor and deterioration of fish products [27]. There was a similar trend in the increase of TBA values in all three groups till the 10th day of salting. The fish muscle is prone to oxidation in the absence of a large amount of salt or starter culture, as observed by the steady increase in TBA values in the NS group. Comparatively low levels of TBA values were detected in the SS group, which is consistent with the results obtained in Harbin dry sausages inoculated with bacterial strains [28]. 

### 3.3. Accumulation of Biogenic Amines during the Processing of Salted Herrings

There is a growing concern about the biogenic amine content in fermented foods due to their ability to cause food poisoning [8]. In this study, we estimated the levels of five biogenic amines: putrescine, cadaverine, histamine, octopamine, and tyramine, in salted herring samples at different processing stages and established the effects of the treatment (evisceration with inoculum) and processing time on biogenic amine accumulations. According to the results of the two-way ANOVA analysis, evisceration with inoculum, processing time, and their interaction all significantly impacted biogenic amine accumulation (Table 1). Cadaverine and putrescine levels increased rapidly, with cadaverine becoming the dominant biogenic amine in the samples. Histamine, octopamine, and tyramine levels were found to be significantly lower. The most common biogenic amines found in high concentrations in fermented dairy products, sausages, and fish products are putrescine and cadaverine. [29]. Significantly elevated levels of cadaverine were found in the TS group. The accumulation of biogenic amine requires amino acids, amino acid decarboxylases-producing bacteria, and favorable conditions for bacterial growth and decarboxylating activity. Essential requirements for cadaverine formation, such as lysine content from muscle protein (data not shown) and abundant enzymes from the visceral microbiota, were present in the TS group.

The FDA recommendation for the limit for histamine is 50 mg/kg [30], and levels lower than that were detected in the TS40, NS20, and NS40 samples. The tyramine concentrations were low (ranging from 0.29 mg/kg to 0.48 mg/kg) in all the samples. Octopamine was detected in the FH group and in the early stages of salting in other groups. 

When compared to the gutted groups, the traditionally processed salting herring had the highest accumulation of biogenic amine. This suggested that microorganisms from the digestive tract play an important role in the formation of biogenic amines. While gutting was an effective method of reducing total biogenic amine content. The addition of autochthonous *S. saprophyticus* (M90–61) in the SS group showed an effective inhibition on histamine accumulation, consistent with other studies where screening potential *Staphylococci* starter cultures helped control biogenic amine in fermented fish and meat products [8,31]. 

### 3.4. Estimation of Volatile Compounds in the Salting Processing

Volatile compounds in different groups of salted herrings were analyzed with the HS-GC-MS and identified through the NIST1.1 mass spectral database. A total of 52 volatile flavor compounds were detected in fresh and salted herring samples, constituting 21 aldehydes, 12 alcohols, 9 ketones, 4 acids, 1 nitrile, and 5 others (Table A2). Based on the two-factor ANOVA analysis, both processing time and technique, as well as their interaction, influenced the volatile component significantly (*p* < 0.05). The volatile component increased after salting, where aldehydes and alcohols, followed by ketones, were dominant in all samples (Figure 2). Variations were present among the processing groups. Nitrile content was significantly (*p* < 0.05) higher in the TS group than in the other two groups, and aldehyde accumulated high, under both traditional processing and with starter addition. 

A total of 21 compounds, constituting 14 aldehydes, 4 alcohols, 2 ketones, and 1 nitrile, were detected with OAV > 1 (Table 2) in the samples. These were considered odor-active compounds and played an important role in flavor development [32]. Different contents of these odor-active compounds were detected in all the samples. Fishy, grassy, fatty, and green were the primary flavors in the salted herrings. 

Due to the low flavor threshold, aldehydes are considered the major contributors to the unique flavor of fermented meat products [6,33]. In the final salted herring products, aldehydes were the most abundant flavor compounds (Figure 3), which is consistent with the study on salted-dried white herring [3]. Hexanal and nonanal were the most abundant aldehydes due to the peroxidation of n-6 polyunsaturated fatty acids, which endowed the product with a fatty, green, floral, and citrus odor (Table 2). Octanal, (E)-2-nonenal, (E)-2-decenal, and 2,6-nonadienal had low thresholds and gave off fatty, fruity, and green flavors (Table 2). Heptanal and (Z)-4-heptenal played significant roles in giving the dry fishy flavor. Similar results were obtained with a previous study on salted-dried fish [3].

Alcohols were the second most abundant component in salted herring, which accumulated due to the β-oxidation of lipids [34]. The highest level of alcohol was observed in the SS40 sample (Figure 3). Most alcohols, especially saturated alcohols, have a high threshold, so they contribute less to flavor development. However, some unsaturated alcohols such as 1-octen-3-ol and 1-penten-3-ol are crucial in flavor development [35]. In salted herring, these alcohols were found in immense proportions in all three groups. 1-octen-3-ol, also known as mushroom alcohol, is the top note in lavender, and it has a low threshold and brings the fatty, fruity, grassy, and mushroom flavors to the final products (Table 2).

Ketones are another set of potent components in salted herring, derived from lipid auto-oxidation or amino acid degradation due to the Strecker reaction [35]. 2,3-Pentanedione and (E, E)-3,5-octadien-2-one are the detected ketones that provide fruity and grassy flavors due to their low threshold.

According to the OVA analysis, the contribution of the acids to the final product was insignificant. They were scantly detected in NS40 and SS40 samples. 3-methyl-butanoic acid is one of the key aroma components in fermented sausage [36], which is also found in NS40 and SS40 (Table A2). 

High Trimethylamine (TMA) levels were detected in all experimental groups, giving the final products a fishy, salty, and deterioration-like aroma (Table A2 and Table 2). This result was consistent with the previous studies on salted-dried white herring and *surströmming*, the traditional Swedish sour herring [3,37]. The TMA content is usually used as a biomedical index for seafood quality, and its accumulation denotes bacterial spoilage [26]. Indigenous microorganisms found in aquatic animals, such as *Aeromonas* spp., *psychrotolerant Enterobacteriaceae*, *P. phosphoreum*, *Shewanella putrefaciens*-like organisms, and *Vibrio* spp., stimulate TMA production from trimethylamine oxide (TMAO) [26]. In this study, the TMA content of the TF group was significantly (*p* < 0.05) higher than the NS and SS groups due to the presence of viscera.

### 3.5. Influence of Different Salting Processes on the Volatile Compounds

The PLS-DA model based on volatile substances was developed to provide a clear profile of the correlation and segregation of flavor composition in each salted herring group. The R^2^X (cum) and R^2^Y (cum) values were 0.954 and 0.915, respectively, according to the results. For the dataset, the predictive power of the Q2 (cum) model was 0.656, indicating that the PLS-DA model was adequate for analysis and prediction. In the Bioplot (Figure 4), the location shift of the samples indicated that the volatile compounds of the salted fish varied based on salting stages and distinct processing methods. Initially, TS5, NS5, and SS5 clustered into the same quadrant, indicating the existence of a similar flavor profile in the early salting stages. Gradually, the diversity of the flavor composition between the TF group and the other two groups became more extensive. Samples from TS40 were distributed in distant quadrants from NS40 and SS40, reflecting the differences among their aroma fingerprints. 

### 3.6. Correlations between the Microbiota and Volatile Components

The development of flavor in the salted herring is a complex process, which is affected by different processing techniques, microbial activities, and endogenous enzymes from the herring’s digestive system. We visually displayed (Figure 5) the correlation between the 10 dominant bacteria genus and volatile components through a cluster correlation heatmap. According to the results, the microorganisms and volatile compounds could be classified into different categories. *Staphyloccocus* showed an outstandingly significant positive correlation (** Rho > 0.6, *p* < 0.01) with 14 aromatic compounds in Cluster II. It demonstrated that the emergence of flavor was one of the essential metabolic attributes of coagulase-negative *Staphylococci* [34]. Autochthonous species such as *S. equorum*, *S. saprophyticus*, *S. xylosus*, and *S. carnosus* were commonly found in fermented sausages, meat, or fish products [38]. They were involved in carbohydrate fermentation, amino acid conversion, lipid β-oxidation, and esterase activities [39]. All these contributed to the accumulation of flavor compounds, such as small peptides, free amino acids, free fatty acids, and volatile substances. Thus, promoting the development of the texture, aroma, and taste of the final products [40]. 

According to the correlation heatmap, *Pseudomonas*, *Acinetobacter*, *Shewanella*, *Flavobacterium*, and *Photobacterium* displayed significantly positive relevance to volatile compounds in Cluster I but negative relevance to a majority of the volatile compounds in Cluster II and weak correlation with flavor compounds in Cluster III. Similarly, *Psychrobacter*, *Cobetia,* and *Pseudoalteromonas* also had a weak influence on aromatic compounds. *Vibrio* exhibited a significantly negative relationship with nine different volatile compounds. *Shewanella* exhibited a significant relationship with trimethylamine (** Rho > 0.7, *p* < 0.01), giving a fishy, salty, and deteriorating flavor to the salted herrings.

Using the multivariate statistical analysis, we attempted to investigate the effects of microbial activities and different treatment methods on product quality. Based on the multiple correlation analysis, we can infer that there is a correlation between changes in microorganisms and volatile substances. During the processing of salted herrings, the accumulation of certain substances could result from interactions between microorganisms, endogenous enzymes, and salt, involving complex biochemical reactions. Therefore, for future research, it is necessary to comprehensively explore the specific metabolic capacity of microorganisms, investigate the changes of endogenous enzymes in products, and analyze the relationship between microbial activities, endogenous enzymes, and the quality of the products.

## 4. Conclusions

According to the data obtained from three salted herring groups, the processing technique directly influenced the microbiota, physical and chemical characteristics, accumulation of biogenic amines, and flavor development. Furthermore, the microbiota is differentiated based on the quality of salted herrings. The diversity of microorganisms and endogenous enzymes in the viscera acted together to form the unique character of salted herrings in the traditional processing group. Traditional processing resulted in an increase in the accumulation of biogenic amines and nitrogenous compounds, and evisceration effectively inhibited this, as seen in the NS and SS groups.

The addition of *Staphylococcus saprophyticus* M90–61 as the starter culture in salted white herring could promote its safety by suppressing the proliferation of other microorganisms, especially spoilage and pathogenic bacteria. The flavor of the salted white herring also could be influenced after the autochthonous strain adding since the *Staphyloccocus* showed a significant positive correlation with 14 aromatic compounds. Further, the suitability of *Staphylococcus saprophyticus* M90–61 in salted white herrings need to be confirmed further with multiple sensory analysis in future work.

## Figures and Tables

**Figure 1 foods-12-00406-f001:**
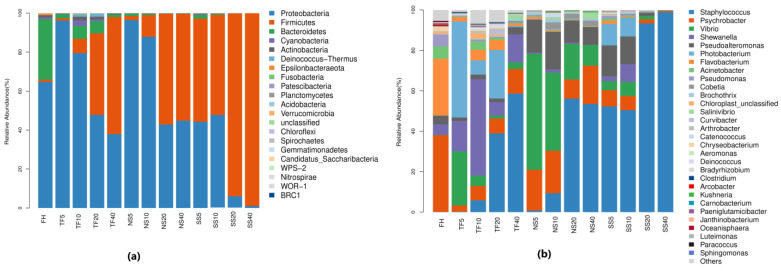
Bacterial community structures. (**a**) Relative abundance of the bacteria of fresh and salted herrings at the phylum level (Top 20 was selected); (**b**) relative abundance of the bacteria of fresh and salted herrings at the genus level (Top 30 was selected).

**Figure 2 foods-12-00406-f002:**
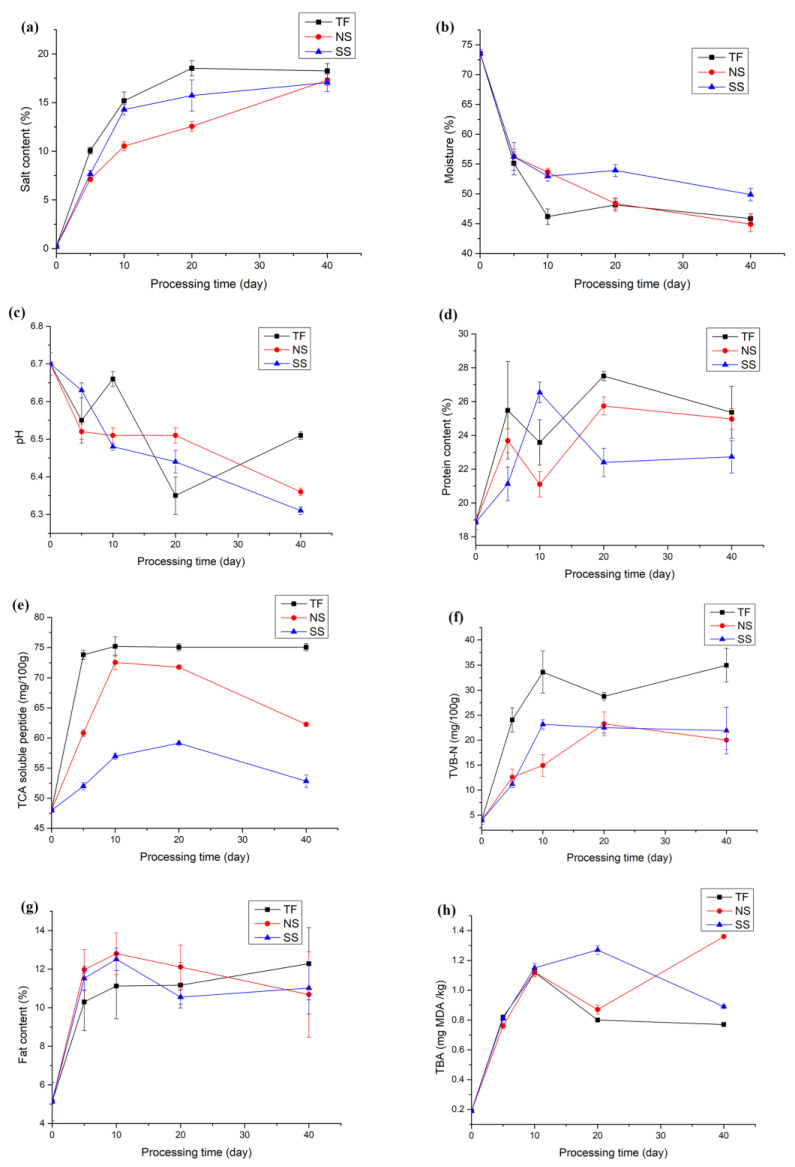
Physicochemical properties of salted herrings. (**a**) Salt content; (**b**) moisture content; (**c**) pH; (**d**) protein content; (**e**) TCA soluble peptide content; (**f**) TVB-N content; (**g**) fat content; and (**h**) TBA values.

**Figure 3 foods-12-00406-f003:**
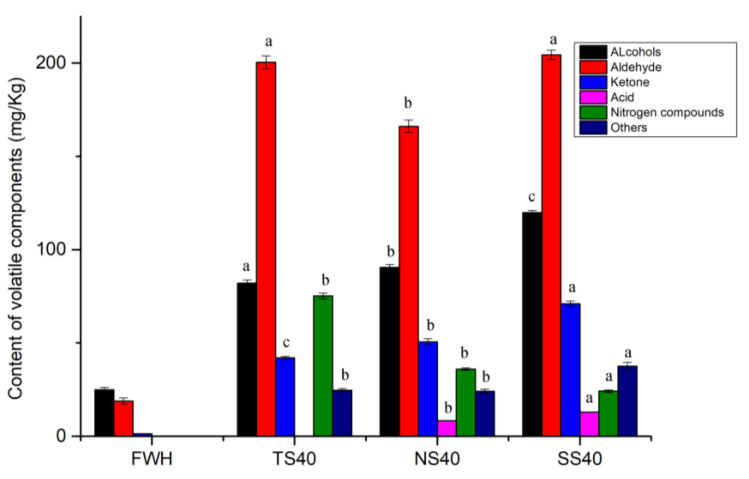
The content of volatile components in the salted herrings processed under different conditions. Lowercase letters (a, b, c) indicate significant differences for evisceration with inoculum for *p* < 0.05.

**Figure 4 foods-12-00406-f004:**
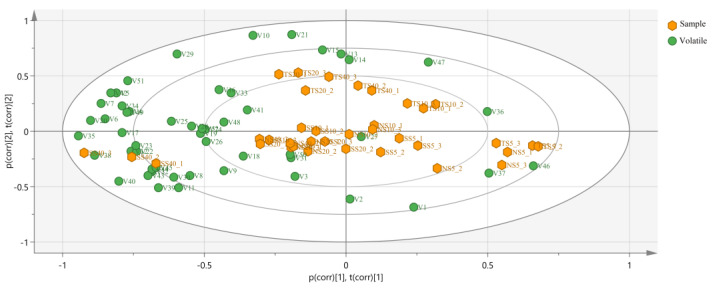
The biplot of volatile flavor compounds correlated with different processing techniques based on OPLS−DA in salted herrings.

**Figure 5 foods-12-00406-f005:**
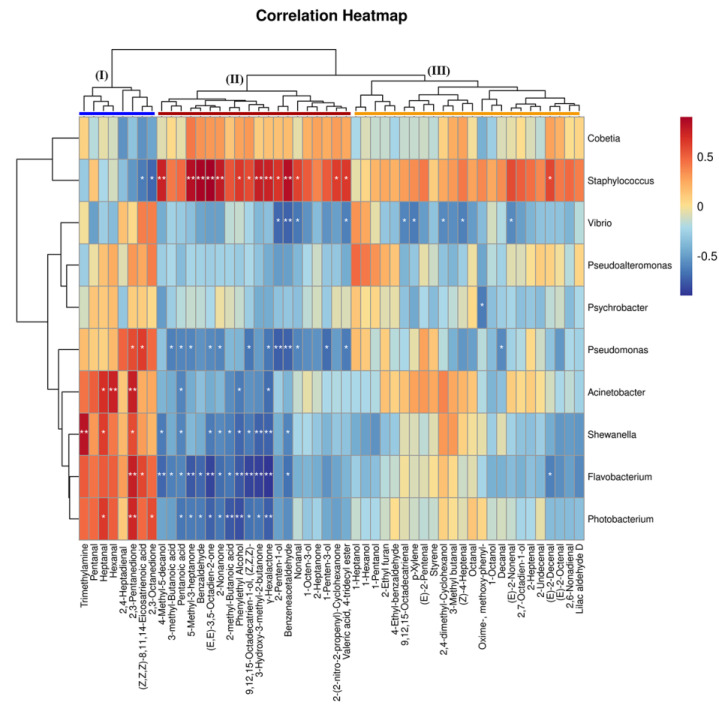
Heatmap of Spearman’s rank correlation between the top 10 genera and volatile compounds in salted herrings. ** Rho > 0.7, *p* < 0.01; * 0.6 < Rho < 0.7, *p* < 0.05.

**Table 1 foods-12-00406-t001:** Biogenic amine contents of salted herring in different salting stages (mg/kg).

BAs	Treatment	Salting Time (day)				Two-Way ANOVA		
		5	10	20	40	P-T	E-I	P-T * E-I
Put	TF	3.82 ± 0.11 ^aD^	2.66 ± 0.09 ^aC^	5.59 ± 0.11 ^aB^	6.43 ± 0.08 ^aA^	**	**	**
	NS	0.60 ± 0.03 ^bD^	2.06 ± 0.09 ^bC^	3.33 ± 0.21 ^bB^	2.05 ± 0.07 ^bA^			
	SS	0.53 ± 0.04 ^cD^	2.24 ± 0.13 ^cC^	1.38 ± 0.07 ^cB^	2.18 ± 0.03 ^cA^			
Cad	TF	72.54 ± 0.70 ^aA^	25.16 ± 0.07 ^aD^	34.26 ± 0.41 ^aC^	67.89 ± 0.09 ^aB^	**	**	**
	NS	ND ^cA^	3.16 ± 0.97 ^cD^	7.32 ± 0.04 ^cC^	4.85 ± 0.11 ^cB^			
	SS	3.89 ± 0.38 ^bA^	11.31 ± 0.35 ^bD^	2.31 ± 0.12 ^bC^	1.70 ± 0.07 ^bB^			
His	TF	ND ^bB^	ND ^bB^	ND ^bB^	1.10 ± 0.04 ^bA^	**	**	**
	NS	ND ^aB^	ND ^aB^	ND ^aB^	1.87 ± 0.13 ^aA^			
	SS	ND ^cB^	ND ^cB^	ND ^cB^	ND ^cB^			
Oct	TF	0.68 ± 0.05 ^cA^	ND ^cB^	ND ^cC^	ND ^cD^	**	**	**
	NS	0.84 ± 0.07 ^aA^	0.63 ± 0.07 ^aB^	0.62 ± 0.05 ^aC^	ND ^aD^			
	SS	0.63 ± 0.05 ^bA^	0.52 ± 0.03 ^bB^	0.39 ± 0.04 ^bC^	ND ^bD^			
Tyr	TF	0.49 ± 0.03 ^bA^	0.27 ± 0.07 ^bA^	0.24 ± 0.06 ^bB^	0.29 ± 0.05 ^bA^	*	*	**
	NS	0.32 ± 0.05 ^aA^	0.45 ± 0.04 ^aA^	0.35 ± 0.04 ^aB^	0.38 ± 0.07 ^aA^			
	SS	0.41 ± 0.03 ^aA^	0.43 ± 0.06 ^aA^	0.34 ± 0.04 ^aB^	0.45 ± 0.09 ^aA^			
Total	TF	77.53 ± 0.79 ^aB^	28.09 ± 0.23 ^aD^	40.09 ± 0.50 ^aC^	75.72 ± 0.16 ^aA^	**	**	**
	NS	1.76 ± 0.05 ^bB^	6.30 ± 0.02 ^bD^	11.62 ± 0.23 ^bC^	9.15 ± 0.15 ^bA^			
	SS	5.46 ± 0.11 ^bB^	14.50 ± 0.48 ^bD^	4.42 ± 0.15 ^bC^	4.34 ± 0.04 ^bA^			

Data were expressed as the mean values with their standard deviations. ND means not detected under the current analysis conditions. P-T means the processing time. E-I means the treatment of evisceration with inoculum. P-T * E-I means their interaction. ** means *p* < 0.001, and * means *p* < 0.01. Lowercase and uppercase superscript letters indicate significant differences for evisceration with inoculum and processing time, respectively, for *p* < 0.05.

**Table 2 foods-12-00406-t002:** The OAV, odor descriptions, and threshold values of volatile components in the salted herring samples.

No.	Compound	Odor Description	Threshold (µg/Kg)	Treatment	Salting Time (day)				Two-Way ANOVA		
					5	10	20	40	P-T	E-I	P-T * E-I
V1	1-Hexanol	Green, grassy	5.60	TF	4.05 ± 0.55 ^aA^	1.73 ± 0.70 ^aB^	ND	0.95 ± 0.84 ^aB^	***	***	***
				NS	21.19 ± 6.84 ^bA^	6.01 ± 0.84 ^bB^	5.77 ± 0.27 ^bB^	5.24 ± 0.37 ^bB^			
				SS	10.95 ± 2.33 ^bA^	8.93 ± 0.62 ^bB^	7.08 ± 0.10 ^bB^	6.67 ± 0.27 ^bB^			
V2	1-Heptanol	Fresh, nutty	5.40	TF	5.19 ± 0.67 ^aA^	3.21 ± 0.21 ^aB^	4.69 ± 0.43 ^aB^	3.89 ± 0.19 ^aB^	***	***	*
				NS	10.19 ± 3.43 ^bA^	5.93 ± 0.19 ^bB^	5.37 ± 0.37 ^bB^	5.12 ± 0.75 ^bB^			
				SS	8.83 ± 0.84 ^bA^	7.10 ± 1.39 ^bB^	4.94 ± 0.75 ^bB^	7.28 ± 0.75 ^bB^			
V5	1-Octen-3-ol	Fishy, grassy	1.50	TF	66.00 ± 6.43 ^bC^	155.11 ± 11.24 ^bB^	215.56 ± 32.46 ^bAB^	168.00 ± 16.34 ^bA^	***	***	***
				NS	113.33 ± 46.77 ^bC^	152.00 ± 11.10 ^bB^	140.67 ± 6.11 ^bAB^	132 ± 17.90 ^bA^			
				SS	110.89 ± 12.39 ^aC^	180.67 ± 17.09 ^aB^	138.00 ± 21.67 ^aAB^	222.89 ± 20.38 ^aA^			
V7	2-Penten-1-ol	Green, plastic	89.20	TF	0.53 ± 0.05 ^bD^	1.14 ± 0.06 ^bC^	1.37 ± 0.13 ^bB^	1.22 ± 0.07 ^bA^	***	***	**
				NS	0.68 ± 0.10 ^bD^	1.11 ± 0.08 ^bC^	1.14 ± 0.06 ^bB^	1.44 ± 0.03 ^bA^			
				SS	0.96 ± 0.14 ^aD^	1.24 ± 0.15 ^aC^	1.37 ± 0.05 ^aB^	1.57 ± 0.05 ^aA^			
V13	Pentanal	Painty, cardboardy	12.00	TF	ND	ND	4.33 ± 0.65 ^aA^	3.92 ± 0.44 ^aAB^	***	***	***
				NS	ND	ND	ND	ND			
				SS	3.19 ± 0.42 ^bB^	ND	ND	ND			
V14	Hexanal	Green, grassy	5.00	TF	56.47 ± 9.07 ^aAB^	89.00 ± 1.40 ^aAB^	135.07 ± 15.13 ^aAB^	99.87 ± 6.99 ^aAB^	ns	***	***
				NS	77.33 ± 10.21 ^cAB^	79.27 ± 9.09 ^cA^	49.80 ± 8.40 ^cAB^	62.07 ± 5.45 ^cA^			
				SS	121.07 ± 13.09 ^bAB^	96.33 ± 13.70 ^bA^	61.93 ± 11.03 ^bAB^	66.53 ± 5.73 ^bA^			
V15	Heptanal	Dry fish	2.80	TF	42.14 ± 12.23 ^aA^	43.33 ± 1.83 ^aA^	66.19 ± 12.44 ^aA^	59.76 ± 3.32 ^aA^	ns	***	***
				NS	37.98 ± 4.23 ^bA^	44.64 ± 0.35 ^bA^	30.24 ± 3.47 ^bA^	41.79 ± 3.52 ^bA^			
				SS	48.57 ± 2.14 ^bA^	49.26 ± 8.66 ^bA^	35.24 ± 5.82 ^bA^	39.29 ± 8.67 ^bA^			
V16	Octanal	Fatty, pungent	0.59	TF	179.66 ± 55.78 ^aB^	164.97 ± 8.70 ^aB^	285.31 ± 49.31 ^aB^	274.01 ± 11.53 ^aA^	*	ns	**
				NS	197.74 ± 40.23 ^aB^	222.03 ± 3.39 ^aB^	157.63 ± 21.23 ^aB^	239.54 ± 17.08 ^aA^			
				SS	230.51 ± 23.91 ^aB^	249.15 ± 69.62 ^aB^	188.70 ± 32.92 ^aB^	248.59 ± 66.22 ^aA^			
V17	Nonanal	Green, fatty	1.10	TF	259.09 ± 121.15 ^bB^	203.33 ± 13.15 ^bB^	340.61 ± 42.16 ^bB^	316.97 ± 13.30 ^bA^	***	**	*
				NS	186.36 ± 9.44 ^bB^	243.94 ± 31.89 ^bB^	292.42 ± 13.18 ^bB^	317.27 ± 31.55 ^bA^			
				SS	290.30 ± 75.71 ^aB^	330.00 ± 92.14 ^aB^	280.30 ± 74.95 ^aB^	488.69 ± 61.18 ^aA^			
V18	Decanal	Fatty	3.00	TF	7.33 ± 2.00 ^aA^	4.22 ± 1.35 ^aA^	7.00 ± 2.19 ^aA^	7.44 ± 1.64 ^aA^	ns	ns	ns
				NS	6.89 ± 0.51 ^aA^	6.22 ± 0.69 ^aA^	7.11 ± 0.38 ^aA^	7.33 ± 1.53 ^aA^			
				SS	6.78 ± 1.39 ^aA^	9.56 ± 6.81 ^aA^	6.78 ± 0.96 ^aA^	9.44 ± 2.36 ^aA^			
V20	Benzeneacetaldehyde	Sweet, flora	6.30	TF	ND	4.07 ± 1.84 ^bB^	9.10 ± 0.40 ^bA^	7.35 ± 0.97 ^bA^	***	***	*
				NS	3.65 ± 2.34 ^bC^	5.50 ± 0.72 ^bB^	7.35 ± 0.24 ^bA^	8.20 ± 2.28 ^bA^			
				SS	4.86 ± 1.95 ^aC^	5.87 ± 0.27 ^aB^	9.74 ± 2.38 ^aB^	11.96 ± 0.97 ^aA^			
V23	(E)-2-Octenal	Green, grassy	3.00	TF	27.78 ± 2.83 ^cB^	38.67 ± 3.61 ^cA^	53.89 ± 2.34 ^cA^	42.44 ± 4.30 ^cA^	*	***	**
				NS	47.44 ± 21.04 ^bB^	51.89 ± 2.17 ^bA^	49.56 ± 2.14 ^bA^	53.22 ± 5.89 ^bA^			
				SS	47.44 ± 2.34 ^aB^	68.33 ± 0.58 ^aA^	51.22 ± 6.05 ^aA^	64.33 ± 0.33 ^aA^			
V25	(E)-2-Nonenal	Fatty, green	0.19	TF	54.38 ± 34.84 ^bA^	57.89 ± 15.79 ^bAB^	105.26 ± 5.26 ^bAB^	84.21 ± 5.26 ^bB^	ns	*	ns
				NS	ND	80.70 ± 24.31 ^bAB^	85.96 ± 8.03 ^bAB^	91.23 ± 53.76 ^bB^			
				SS	117.54 ± 48.90 ^aA^	115.78 ± 77.53 ^aAB^	73.68 ± 22.94 ^aAB^	128.07 ± 34.24 ^aB^			
V26	(E)-2-Decenal	Fruity	0.40	TF	ND	28.33 ± 3.82 ^bA^	42.50 ± 2.50 ^bA^	33.33 ± 3.81 ^bA^	ns	*	ns
				NS	ND	44.17 ± 18.43 ^abA^	47.50 ± 6.61 ^abA^	74.17 ± 70.10 ^abA^			
				SS	70.83 ± 55.92 ^aA^	101.67 ± 108.98 ^aA^	49.17 ± 9.46 ^aA^	68.33 ± 23.76 ^aA^			
V27	2,4-Heptadienal	Fatty, fishy	15.40	TF	2.08 ± 0.45 ^aB^	4.46 ± 0.38 ^aA^	1.26 ± 0.43 ^aC^	0.88 ± 0.32 ^aBC^	**	**	***
				NS	1.69 ± 0.36 ^bB^	1.36 ± 0.51 ^bA^	1.80 ± 0.64 ^bC^	1.00 ± 0.31 ^bBC^			
				SS	1.80 ± 0.99 ^bB^	1.32 ± 0.27 ^bA^	0.99 ± 0.37 ^bC^	2.84 ± 0.21 ^bBC^			
V29	(Z)-4-Heptenal	Fishy, fish oil like	4.20	TF	6.35 ± 0.77 ^aC^	18.49 ± 1.22 ^aB^	29.16 ± 2.68 ^aAB^	27.78 ± 2.08 ^aA^	***	**	***
				NS	10.48 ± 5.04 ^bC^	16.11 ± 2.79 ^bB^	14.84 ± 4.06 ^bAB^	21.27 ± 2.29 ^bA^			
				SS	14.60 ± 4.29 ^aC^	23.17 ± 2.36 ^aB^	17.70 ± 2.21 ^aAB^	20.87 ± 3.47 ^aA^			
V31	2-Undecenal	Soapy, metallic	1.40	TF	ND	ND	ND	ND	ns	***	ns
				NS	ND	4.05 ± 3.93 ^bA^	ND	ND			
				SS	27.38 ± 1.48 ^aA^	24.05 ± 27.04 ^aA^	6.43 ± 1.89 ^aA^	10.00 ± 3.27 ^aA^			
V32	2,6-Nonadienal	Wax, green	0.50	TF	33.33 ± 8.08 ^cC^	45.33 ± 1.15 ^cAB^	68.00 ± 6.00 ^cA^	ND	**	***	***
				NS	37.33 ± 11.02 ^bC^	60.00 ± 17.32 ^bAB^	72.00 ± 4.00 ^bA^	52.00 ± 8.72 ^bBC^			
				SS	56.67 ± 18.58 ^aC^	62.00 ± 2.00 ^aAB^	60.67 ± 25.48 ^aA^	104.67 ± 13.32 ^aBC^			
V36	2,3-Pentanedione	Fruity	30.00	TF	3.14 ± 0.38 ^aA^	4.20 ± 0.79 ^aAB^	3.66 ± 0.34 ^aBC^	3.15 ± 0.30 ^aC^	***	ns	**
				NS	4.12 ± 1.13 ^aA^	3.08 ± 0.39 ^aAB^	2.61 ± 0.59 ^aBC^	3.00 ± 0.64 ^aC^			
				SS	4.79 ± 0.45 ^aA^	3.63 ± 0.42 ^aAB^	3.10 ± 0.38 ^aBC^	2.07 ± 0.12 ^aC^			
V38	(E,E)-3,5-Octadien-2-one	Fruity, grassy	150.00	TF	0.29 ± 0.23 ^cD^	0.60 ± 0.06 ^cC^	0.67 ± 0.09 ^cB^	1.03 ± 0.14 ^cA^	***	***	**
				NS	0.44 ± 0.17 ^bD^	0.64 ± 0.06 ^bC^	1.12 ± 0.06 ^bB^	1.18 ± 0.17 ^bA^			
				SS	0.99 ± 0.13 ^aD^	1.03 ± 0.03 ^aC^	1.07 ± 0.34 ^aB^	2.04 ± 0.10 ^aA^			
V47	Trimethylamine	Fishy, salty, deterioration	8.00	TF	100.88 ± 25.74 ^aB^	142.29 ± 19.09 ^aA^	79.55 ± 2.50 ^aA^	93.92 ± 6.94 ^aB^	***	***	***
				NS	13.04 ± 1.05 ^bB^	43.17 ± 15.63 ^bA^	66.26 ± 7.50 ^bA^	44.92 ± 4.54 ^bB^			
				SS	36.71 ± 1.37 ^bB^	61.42 ± 12.88 ^bA^	40.33 ± 5.19 ^bA^	30.25 ± 8.94 ^bB^			

Data were expressed as the mean values with their standard deviations. ND means not detected under the current analysis conditions. P-T means the processing time. E-I means the treatment of evisceration with inoculum. P-T * E-I means their interaction. *** means *p* < 0.001, ** means *p* < 0.01, * means *p* < 0.05, and ns means *p* > 0.05. Lowercase and uppercase superscript letters indicate significant differences for evisceration with inoculum and processing time, respectively, for *p* < 0.05.

## Data Availability

Data is contained within the article.

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
