# Peer review of "Role of Microorganisms in the Development of Quality during the Fermentation of Salted White Herring (Ilisha elongata)"

_foods, 2023, doi:10.3390/foods12020406_

Round 1

Reviewer 1 Report

In the present paper ‘Role of Microorganisms in the Development of Quality during the Fermentation of Salted White Herring (Ilisha Elongata)’, the authors investigate the role of microorganisms in the quality formation of the salted herring using three groups for different processes of salting herrings. Staphylococcus spp. proved to be the predominant microorganism in each processing group, while a strong positive correlation between Staphyloccocus and 14 aromatic compounds was observed. The manuscript represents interesting data regarding the important of the fish processing technique which directly influence the microbiota, physical and chemical characteristics, accumulation of biogenic amines, and flavor development.

The manuscript is very well written, with many explanations and correlation between the obtained data, the justifications being each time accompanied by appropriate references.

Author Response

Dear reviewer,

Thanks for your earnest review and we carefully modified the English writting under the help of native speakers.

Jiajia Wu, Haiping Mao and Zhiyuan Dai

Reviewer 2 Report

This work presents a piece of very interesting information about the effect of different salting processes on the development of different kinds of microorganisms. All the section of the study was clearly established, and the results were discussed according to previous work. The relevant information, besides the main microorganism developed in each salted treatment evaluated, was the correlation between volatile components and the microbiota. I do not have any observations or comments. 

Author Response

Dear reviewer,

We appreciate for your earnest review and we modified the manuscript again, especially the English writting.

Jiajia Wu, Haiping Mao, Zhiyuan Dai

Reviewer 3 Report

Dear Authors,

The original study is interesting and applies an important advance in the evaluation of autochthonous microorganism activity and effect for fermented food production. In general, the study is fairly designed, but some improvements are necessary to correct some passages (especially language), justify the choice for the selected strain, and revise some core analysis of results.

Introduction could be improved to include a paragraph about the Staphylococcus saprophyticus M90–61 (maybe after Line 54). Since this strain is an interesting candidate to produce fermented salted white herring, more information has to be included to justify it. My suggestion is the inclusion of growth characteristics (optimum temperature, maximum saltiness), technological properties (enzymes), and safety aspects (production of toxins), other studies reporting the natural presence of this strain…

Line 31: maybe “would preserve the harvested white herring by heavy salting it.”

Line 38: bamboo stick is impaled into…

Line 41: mainly define the characteristics…

Line 51: bacteria, pathogenic microorganisms, or…

Line 77: Please revise the use of “procession”

Line 81: Samples from each group were picked…

Line 114: fat content, respectively, according to the AOAC

Line 118: 5 min…

Statistical analysis is partially deficient and must be revised. Two independent factors are being considered: evisceration with inoculum (TF, NS, and SS) and processing time (5, 10, 20, and 40 days). A One-way ANOVA is not suitable for this design and Sections 3.2, 3.3, and 3.4 must be revised. A GLM or Two-way ANOVA could be used for data analysis, please indicate the method in Section 2.6. Once the results of statistical analysis are obtained, please indicate them in Figures 2 and 3 and Tables 1 and 2 and revise sections 3.2, 3.3, and 3.4.

Line 198: produced unpleasant odor…

Figure 4: Please indicate each group with a different color. The use of one single color largely limits the visualization of different groups in this figure.

Line 350: herring is a complex process…

Conclusion: More focus should be given to the autochthonous strain. Please include a statement indicating the which nutritional and technological properties were affected by the autochthonous strain (Line 394).

Author Response

Dear reviewers and editors,

We appreciate a lot for all of your conscientious and objective comments. According to reviewer 3’s suggestions, we revised the English writing of the manuscript and the modifications are marked out using the “Track Changes” function. The other parts were modified as follows and marked with highlight brightness.

Suggestion1: Introduction could be improved to include a paragraph about the Staphylococcus saprophyticus M90–61 (maybe after Line 54). Since this strain is an interesting candidate to produce fermented salted white herring, more information has to be included to justify it. My suggestion is the inclusion of growth characteristics (optimum temperature, maximum saltiness), technological properties (enzymes), and safety aspects (production of toxins), other studies reporting the natural presence of this strain…

Response1: We added on paragraph about the Staphylococcus saprophyticus M90–61in the Introduction part. The data related to its growth characteristics, technological properties and safety aspects in this additional paragraph has not been published yet.

Suggetion2: Statistical analysis is partially deficient and must be revised. Two independent factors are being considered: evisceration with inoculum (TF, NS, and SS) and processing time (5, 10, 20, and 40 days). A One-way ANOVA is not suitable for this design and Sections 3.2, 3.3, and 3.4 must be revised. A GLM or Two-way ANOVA could be used for data analysis, please indicate the method in Section 2.6. Once the results of statistical analysis are obtained, please indicate them in Figures 2 and 3 and Tables 1 and 2 and revise sections 3.2, 3.3, and 3.4.

Response2: We applied the 2-ways ANOVA to analyzed the data in Sections 3.2, 3.3, and 3.4, as evisceration with inoculum and processing time were considered as two independent factors, and the method is indicated in Section 2.6. In this case, we reorganized to Table 1 and Table 2, and modified the Figure 3. For Figure 2, we added the corresponding 2-way ANOVA analysis data in Appendix Table A3. Furthermore, some of result explanations about the results in sections 3.2, 3.3, and 3.4 were improved.

Suggestion 3: Some improvements are necessary to correct some passages.

Line 31: maybe “would preserve the harvested white herring by heavy salting it.” ; Line 38: bamboo stick is impaled into…; Line 41: mainly define the characteristics…; Line 51: bacteria, pathogenic microorganisms, or…; Line 77: Please revise the use of “procession”; Line 81: Samples from each group were picked…; Line 114: fat content, respectively, according to the AOAC; Line 118: 5 min…; Line 198: produced unpleasant odor…;Line 350: herring is a complex process…

Response3: We apologize for the careless errors we made and have modified all of those errors mentioned above. Furthermore, other corrections were also marked out.

Suggestion 4: Figure 4: Please indicate each group with a different color. The use of one single color largely limits the visualization of different groups in this figure.

Response4: Here we appreciate your reasonable suggestion again, but we failed to indicate each group with a different color in the Simca software. I feel regret to make no improvement on this point.

Suggestion 5: More focus should be given to the autochthonous strain. Please include a statement indicating the which nutritional and technological properties were affected by the autochthonous strain (Line 394).

Response5: One single paragraph was carried out the state the influence of the Staphylococcus saprophyticus M90–61 on the microbiota and flavor of the salted herring.

We added the corresponding author’s institutional email in the title page. [email protected].

Thanks for all of your patient reviews on the manuscript.

Jiajia Wu, Haiping Mao, Zhiyuan Dai.
